# Oxidative Stress Markers in Human Brain and Placenta May Reveal the Timing of Hypoxic-Ischemic Injury: Evidence from an Immunohistochemical Study

**DOI:** 10.3390/ijms241512221

**Published:** 2023-07-30

**Authors:** Benedetta Baldari, Stefania De Simone, Luigi Cipolloni, Paolo Frisoni, Letizia Alfieri, Stefano D’Errico, Vittorio Fineschi, Emanuela Turillazzi, Pantaleo Greco, Amerigo Vitagliano, Gennaro Scutiero, Margherita Neri

**Affiliations:** 1Department of Anatomical, Histological, Forensic and Orthopaedic Sciences, Sapienza University of Rome, Viale Regina Elena 336, 00161 Rome, Italy; benedetta.baldari@uniroma1.it (B.B.); vittorio.fineschi@uniroma1.it (V.F.); 2Department of Clinical and Experimental Medicine, Section of Legal Medicine, University of Foggia, Viale Europa 12, 71122 Foggia, Italy; stefania.desimone@unifg.it (S.D.S.); luigi.cipolloni@unifg.it (L.C.); 3Unit of Legal Medicine, Azienda USL di Ferrara, Via Arturo Cassoli 30, 44121 Ferrara, Italy; paolo.frisoni@ausl.fe.it; 4Department of Medical Sciences, Section of Legal Medicine University of Ferrara, Via Fossato di Mortara 70, 44121 Ferrara, Italy; letizia.alfieri@unife.it; 5Department of Medicine, Surgery and Health, University of Trieste, Strada di Fiume 447, 34149 Trieste, Italy; sderrico@units.it; 6Department of Surgical Pathology, Medical, Molecular and Critical Area, Institute of Legal Medicine, University of Pisa, Via Roma, 55/57, 56126 Pisa, Italy; emanuela.turillazzi@unipi.it; 7Department of Medical Sciences, Section of Obstetrics and Gynecology, University of Ferrara, Via Aldo Moro 8, 44124 Ferrara, Italy; pantaleo.greco@unife.it (P.G.); g.scutiero@ospfe.it (G.S.); 81st Unit of Obstetrics and Gynecology, Department of Biomedical and Human Oncological Science (DIMO), University of Bari, Policlinico, Piazza Giulio Cesare, 11, 70124 Bari, Italy; amerigo.vitagliano@gmail.com

**Keywords:** reactive oxygen species, hypoxic-ischemic brain injury, placenta, immunohistochemical markers, forensic pathology

## Abstract

During pregnancy, reactive oxygen species (ROS) serve as crucial signaling molecules for fetoplacental circulatory physiology. Oxidative stress is thought to sustain the pathogenesis and progression of hypoxic-ischemic encephalopathy (HIE). A retrospective study was performed on the brains and placentas of fetuses and newborns between 36–42 weeks of gestation (Group_1: Fetal intrauterine deaths, Group_2: Intrapartum deaths, Group_3: Post-partum deaths, Control group sudden neonatal death); all groups were further divided into two subgroups (Subgroup_B [brain] and Subgroup_P [placenta]), and the study was conducted through the immunohistochemical investigations of markers of oxidative stress (NOX2, 8-OHdG, NT, iNOS), IL-6, and only on the brain samples, AQP4. The results for the brain samples suggest that NOX2, 8-OHdG, NT, iNOS, and IL-6 were statistically significantly expressed above the controls. iNOS was more expressed in the fetal intrauterine death (Group_1) and less expressed in post-partum death (Group_3), while in intrapartum death (Group_2), the immunoreactivity was very low. IL-6 showed the highest expression in the brain cortex of the fetal intrauterine death (Group_1), while intrapartum death (Group_2) and post-partum death (Group_3) showed weak immunoreactivity. Post-partum death (Group_3) placentas showed the highest immunoreactivity to NOX2, which was almost double that of the fetal intrauterine death (Group_1) and intrapartum death (Group_2) placentas. Placental tissues of fetal intrauterine death (Group_1) and intrapartum death (Group_2) showed higher expression of iNOS than post-partum death (Group_3), while the IL-6 expression was higher in the fetal intrauterine death (Group_1) than the post-partum death (Group_3). The AQP4 was discarded as a possible marker because the immunohistochemical reaction in the three groups of cases and the control group was negative. The goal of this study, from the point of view of forensic pathology, is to provide scientific evidence in cases of medical liability in the Obstetric field to support the clinical data of the timing of HIE.

## 1. Introduction

Reactive oxygen species (ROS) are physiologically produced from cell metabolism and play a dual role as both beneficial and toxic compounds [1,2]. At physiological concentrations, ROS play a role in the defense against pathogens, cell signaling, and mitogenic responses [1,3]. At higher concentrations (e.g., in case of either overload of ROS or depletion of natural antioxidants), ROS cause oxidative stress (OS), potentially leading to impaired cell function and decreased cell viability due to damage to proteins, DNA, and cellular lipids [4,5,6,7].

ROS serve as crucial signaling molecules during pregnancy for fetoplacental circulatory physiology [8,9]. In the first trimester, OS drives the expression of vascular endothelial growth factor (VEGF) through hypoxia-inducible factor-1, thus stimulating placental vascular development [9,10]. Furthermore, during the second and third trimesters, OS may stimulate the overexpression of apoptotic regulators at the placental level, potentially leading to massive apoptosis and placental insufficiency [11,12]. For these reasons, OS was supposed to play an active role in the development of different severe placenta-related obstetrical complications, including pre-eclampsia and intrauterine growth restriction [9,10,11,12]. Additionally, on the fetal side, OS may sustain the pathogenesis and progression of hypoxic-ischemic encephalopathy (HIE), as developing neurons are very susceptible to ROS due to high oxygen consumption, lipid-rich content, and relatively low antioxidant defense [13,14].

HIE represents one of the major causes of neonatal death and neurological disability, with an incidence of 1.5 cases per 1000 live births in developed countries and of 10–20 per 1000 live births in low-middle-income countries [15,16,17]. The causes of HIE are many, including umbilical cord knotting, umbilical cord prolapse, shoulder dystocia, placental abruption, and chronic maternal hypoxia [18,19]. The mechanism of damage from hypoxic-ischemic insult involves a series of events, with a dramatic increase in ROS occurring mainly during the reperfusion and reoxygenation phase (during the first 12–48 h after the hypoxic-ischemic insult) [20,21].

In clinical practice, identifying the timing of hypoxic-ischemic insult underlying HIE is challenging for forensic pathologists [22]. It is often difficult to resolve doubts about potential clinical liability in cases of HIE. In this respect, we hypothesized that the measurement of OS-indicative markers in the human brain and placenta could be helpful in the identification of the timing of the hypoxic-ischemic insult. In order to test our hypothesis, we retrospectively investigated oxidative stress processes by evaluating immunohistochemistry samples from forensic autopsies of fetuses/newborns in which the timing of hypoxic-ischemic insult was unknown (i.e., intrauterine deaths, intrapartum deaths, and post-partum deaths), as compared with a control group in which the timing of insult was established (sudden death due to acute events).

The reason for our study is to provide scientific evidence in cases of medical liability in the obstetric field to support the clinical data on the timing of HIE.

For the identification of the timing of the hypoxic-ischemic insult, we investigated the potential usefulness of assessing the expression of specific OS markers NADPH oxidase 2 (NOX2), 8-hydroxy-2′-deoxyguanosine (8-OHdG), nitrotyrosine (NT), nitric oxide synthase (iNOS) and Interleukin 6 (IL-6), both in cortex brain and placenta samples, only in the brain we studied the immunohistochemical expression of Aquaporin 4 (AQP4).

## 2. Results

A preliminary semiquantitative analysis was performed, both for brain and placenta samples; the markers investigated were NADPH oxidase 2 (NOX2), 8-hydroxy-2′-deoxyguanosine (8-OHdG), nitrotyrosine (NT), nitric oxide synthase (iNOS), and Interleukin 6 (IL-6). Based on the results of the semi-quantitative analysis, the quantitative analysis was performed using ImageJ software (see Section 4. Materials and Methods and Section 2.1 and Section 2.2 of Results).

The immunohistochemical investigation of Aquaporin 4 (AQP4) was not significant because the marker reaction in the Group_1 B, Group_2 B, Group_3 B, and Controls B was negative, so after a preliminary semiquantitative analysis, the AQP4 was discarded as a possible marker.

### 2.1. Brain Tissue

The quantitative expression of oxidative stress markers, NOX2, 8-OHdG, NT, iNOS, and IL-6, within each group (Group_1 B: Fetal intrauterine deaths, Group_2 B: Intrapartum deaths, Group_3 B: Post-partum deaths, Controls B) is reported in Table 1.

The expression of NOX2, 8-OHdG, NT, iNOS, and IL-6 was significantly different between study groups (*p*-values 0.009, 0.003, 0.005, <0.001, and 0.001, respectively). Notably, all those markers were more highly expressed in cases (Group_1 B, Group_2 B, and Group_3 B) compared with Controls B (except for iNOS in Group_2 B vs. Controls B; *p*-value > 0.05).

Individual comparisons between Group_1 B, Group_2 B, and Group_3 B found no significant differences in terms of NOX2, 8-OHdG, and NT expression. Differently, regarding iNOS and IL-6, the highest expression was found in Group_1 B (*p* < 0.05 compared to Group_2 B and Group_3 B) (Figure 1 and Figure 2).

Moreover, Group_3 B showed higher immunoreactivity to iNOS than Group_2 B (*p* < 0.05). Box plots of the immunohistochemical expression of NOX2, 8OHdG, NT, iNOS, and IL-6 in the brain cortex of each group are shown in Appendix A.

### 2.2. Placental Tissue

The quantitative expression of oxidative stress markers in placental tissue within groups (Group_1 P: Fetal intrauterine deaths, Group_2 P: Intrapartum deaths, Group_3 P: Post-partum deaths, Controls P) is reported in Table 2.

The expression of NOX2, NT, iNOS, and IL-6 differed significantly between groups (*p*-values 0.01, 0.019, <0.001, and 0.026, respectively), while 8-OHdG positivity was similar (*p* = ns).

Regarding NOX2 immunohistochemistry, the highest expression was found in Group_3 P (i.e., significantly higher compared to Group_1 P, Group_2 P, and Controls P; all *p* < 0.05) (Figure 3).

Moreover, Goup_2 P and Group_3 P showed significantly higher positivity to NT compared to controls (*p* < 0.05), while no difference in NT immunoreactivity between Group_1 P, Group_2 P, and Group_3 P was found (*p* = ns).

The highest expression of iNOS was observed in Group_1 P and Group_2 P (i.e., significantly higher compared to Group_3 P and Controls P; all *p* < 0.05). Nevertheless, the comparison between Group_1 P and Group_2 P did not find significant differences in iNOS positivity (Figure 4).

Finally, the expression of IL-6 was higher in Group_1 P than in Group_3 P. No other statistical differences were found. Box plots of the immunohistochemical expression of NOX2, 8-OHdG, NT, iNOS, and IL-6 in the brain cortex of each group are shown in Appendix A.

## 3. Discussion

In forensic medicine, the identification of the timing of hypoxic-ischemic injury underlying the HIE remains a challenge. The autopsy alone is often insufficient to answer all doubts during forensic examinations, leading to the potential misinterpretation of unrecognized chronic fetal problems as the consequence of acute perinatal injuries. Therefore, it is often difficult to resolve doubts about potential clinical liability in cases of HIE [22,23].

After a hypoxic-ischemic attack (i.e., impaired cerebral blood flow and oxygen delivery to the brain), the pathologic events of HIE involve the generation of OS, with the release of oxygen and nitrogen species, calcium overloading, ROS generation, ionic imbalance, inflammation, apoptosis, autophagy, and necrosis [24,25].

HIE occurs in two phases (i.e., primary energy failure and secondary energy failure), where the second phase is characterized by massive ROS release 6–48 h after the initial injury, resulting in damage to neuronal cell membranes, necrosis, and apoptosis [26,27,28,29]. In this pilot study, we investigated the potential usefulness of assessing the expression of specific OS markers by immunohistochemistry (i.e., NOX2, 8-OHdG, iNOS, NT, and IL-6) in fetal/neonatal brain cortex and placenta for the identification of the timing of the hypoxic-ischemic insult. Previous in vitro studies about the edema that occurs during the early stage of ischemic stroke and brain trauma show the role of AQP4. The HIE is characterized by brain edema, so we tested AQP4 in our three groups and controls of newborn and fetus brain samples. The immunohistochemical analysis of AQP4 was not significant, so the AQP4 was discarded as a possible marker in HIE. This result suggests that the not complete maturation of brain tissue in fetuses and newborns could be the cause.

On the other hand, NOX2, 8-OHdG, NT, iNOS, and IL-6 were generally more highly expressed in the brain cortex of three groups of cases (fetal intrauterine death, intrapartum death, and post-partum death) compared with controls. Interestingly, the widest differences between the three groups, fetal intrauterine death, intrapartum death, and post-partum death, were found in terms of iNOS and IL-6 expression. Regarding iNOS, the highest expression in brain tissue was observed in fetal intrauterine death (Group_1 B), followed by post-partum death (Group_3 B), while in intrapartum death (Group_2 B), the iNOS immunoreactivity was very low. NOS is an enzyme that catalyzes the production of nitric oxide from l-arginine [30,31]. After asphyxia, nitric oxide generates toxic peroxynitrite (by reacting with superoxide), setting a pre-apoptotic process and resulting in neuronal loss [32,33]. NOS exists in three principal isoforms: endothelial (eNOS), neuronal (nNOS), and inducible NOS (iNOS). nNOS and iNOS are mainly associated with deleterious effects on the brain [31,33,34]. Specifically, murine experiments showed that nNOS is activated immediately after reperfusion, while iNOS is upregulated several hours after the asphyctic injury [34]. This notion is consistent with our finding about the highest expression of iNOS in the cortex of those babies undergoing intrauterine death. Regarding IL-6, fetal intrauterine death (Group_1 B) showed the highest expression in the brain cortex, while intrapartum death (Group_2 B) and post-partum death (Group_3 B) showed weak and similar immunoreactivity. IL-6 is a cytokine with pleiotropic activity produced by different brain cells, mainly by astrocytes [35,36,37]. Rapid increases in the levels of IL-6 can exacerbate an ischemic injury by stimulating the apoptosis of neuronal cells [38,39]. Interestingly, Yoon et al. previously found that IL-6 expression was upregulated in the brain of those babies who died due to chorioamnionitis [40]. In our study, the majority of intrauterine death was of unexplained etiology, likewise in most cases reported in the literature. As IL-6 was maximally up-regulated in Group_1 B (intrauterine death), a potential etiological correlation between intrauterine death and infection may not be excluded.

Therefore, those differences in the immunoreactivity of brain tissue to iNOS and IL-6 between fetal intrauterine death (Group_1 B) (i.e., strong to iNOS and IL-6), intrapartum death (Group_2 B) (low to iNOS and IL-6) and post-partum death (Group_3 B) (low to iNOS and moderate to IL-6) may represent a first step in identifying the timing of lethal injuries for forensic purposes.

In addition to the abovementioned findings on the brain cortex, we found a different expression of NOX2, NT, iNOS, and IL-6 between groups at the placental level, while 8-OHdG positivity was similar. Post-partum death (Group_3 P) placentas showed the highest immunoreactivity to NOX2, which was almost double that of fetal intrauterine death (Group_1 P) and intrapartum death (Group_2 P) placentas. NOX2 is a membrane-bound enzyme catalyzing the production of a superoxide free radical by transferring one electron to oxygen from NADPH [41,42]. Its activity is predominant in the apical microvillous membrane of the syncytiotrophoblast in the term placenta [43,44]. However, the physiological role of this enzyme at the maternal–fetal interface in term pregnancies is still unclear. In a previous study, Manes [45] suggested that NOX2 might respond as an oxygen-sensing mechanism and a signaling molecule at the trophoblast surface. In line with this latter speculation, Matsubara and Sato [46] found a dramatic increase in NOX2 expression at term in placentas with signs of vasculopathy. Concerning our findings, we may hypothesize that the impressive NOX2 upregulation found in post-partum death (Group_3 P) may be due to altered placental blood flow during labor. This result was consistent with the higher (but not significant) NT expression in intrapartum death (Group_2 P) and post-partum death (Group_3 P) placentas compared to fetal intrauterine death (Group_1 P). Notably, NT can be considered as an index of oxidative stress arising from peroxynitrite formation and action, which is found in the placenta after vascular damage [47,48]. Thus, both NOX2 and NT may virtually indicate a pathological response of placental tissue against demands for altered blood flow during labor, such as commonly observed in pregnancies complicated by pre-eclampsia or diabetes [49,50].

Finally, placental tissues of fetal intrauterine death (Group_1 P) and in intrapartum death (Group_2 P) showed higher expression of iNOS than post-partum death (Group_3 P), while the IL-6 expression was higher in fetal intrauterine death (Group_1 P) than post-partum death (Group_3 P). These differences in the placental expression of iNOS and IL-6 between groups almost confirmed our findings observed in the brain cortex.

The results of this study demonstrated that the immunohistochemical expression of specific OS markers in the brain cortex and placentas is different between fetuses/newborns undergoing intrauterine, intrapartum, and post-partum death, providing useful information for identifying the timing of the development of HIE.

The first point for the forensic pathologist is to identify the stage of labor or delivery in which the fetus or newborn died. The tools utilized until now are focused on the study of the lungs to determine whether the fetus had breathed or not before death.

The timing of the injury is the key point to demonstrate that the injury occurred at a different time than the strict intrapartum period. The clinical presentation and neuroimaging of hypoxic-ischemic encephalopathy cannot provide an accurate temporal relationship between the event and the pathologic manifestation. The autopsy cannot provide all the necessary information [51].

In cases of fetal or newborn death during labor or in the last part of pregnancy during medical activity could result in a penal and/or civil suit against the doctors or the hospital, according to the Country Law. The expert witness (e.g., in Italy, a forensic pathologist and an obstetrician) must appoint very qualified personnel to perform the autopsy on the fetus or newborn and study the clinical documentation in depth for the evaluation of the conduct of the doctors who carried out the birth and followed the pregnancy. The evaluation must be scrupulous and supported by the literature, so it is crucial to provide further scientific evidence for the evaluation of the cases using immunohistochemical markers.

Forensic pathology aims to implement immunohistochemical assays to corroborate qualitative evidence with semi-quantitative parameters [52]. The importance of this study is the individuation of markers useful to support the clinical data in cases of medical malpractice in the field of obstetrics.

## 4. Materials and Methods

### 4.1. Study Design

This was a retrospective case-control study on fetal/neonatal brains and placental samples collected at two centers (Department of Legal Medicine of Policlinic Umberto I—Sapienza; Department of Legal Medicine of the University of Foggia) from January 2012 to December 2022. The study was designed to investigate whether the use of immunohistochemistry may help establish the timing of hypoxic-ischemic insult in post-mortem fetuses/newborns.

### 4.2. Subjects and Data Collection

Among cases (n = 25), we analyzed the brains and placentas of babies (fetuses/newborns) born between 36–42 weeks of gestation. One sample of brain and placenta was collected for each case. The study involved three case groups (Group_1: Fetal intrauterine deaths [n = 9], Group_2: Intrapartum deaths [n = 9], Group_3: Post-partum deaths [n = 7]). As a control group (Controls), we evaluated the brains and placentas of babies who had undergone sudden neonatal death (n = 6 cases). In order to simplify the report of each comparison between placentas and brains pertaining to the study groups, all groups were further divided into two subgroups (Subgroup_B [brain] and Subgroup_P [placenta]).

For each baby, we collected data about gestational age at delivery, type of delivery, birth weight, survival time after birth, APGAR score, the potential cause of death (as diagnosed by a forensic pathologist), and the timing of death.

All the characteristics of the study subjects are summarized in Table 3.

### 4.3. Methods

All samples of the brain cortex and placentas were evaluated through immunohistochemical investigations for NADPH Oxidase 2 (NOX2), 8-Hydroxy-2′-deoxyGuanosine (8-OHdG), NitroTyrosine (NT), inducible Nitric Oxide Synthase (iNOS) and Interleukin-6 (IL-6). For brain sections, we also performed immunohistochemical analysis for Aquaporin-4 (AQP4) in order to estimate the development of cerebral edema. For each case, sections of about 4 μm thickness were cut; after the hydration, the slices were pretreated for antigen retrieval and then incubated with primary antibody according to the dilution indicated in Table 4. The utilized detection system (CTS005 HRP-DAB system R&D kit, R&D systems, Inc., Minneapolis, MN, USA) was a refined avidin–biotin system in which a biotinylated secondary antibody reacts with several peroxidize-conjugated streptavidin molecules. The positive reaction was visualized by 3,3′-diaminobenzidine (DAB) peroxidation, according to standard methods.

We evaluated the presence of non-specific markings due to the avidin-biotin system. Hence, we carried out tests using a polymer system (BioCare Goat-on-rodent HRP-Polymer), obtaining the markings of the same areas.

Sections were counterstained with hematoxylin, dehydrated, cover-slipped, and observed in a Nikon Eclipse E600 microscope (Nikon, Tokyo, Japan). Quantification of NOX2, 8-OHdG, NT, iNOS, and IL-6 positive-stained cells was performed by the ImageJ software (imagej.nih.gov/ij/), as previously described [53,54], using the “Manual Cell Counting and Marking” protocol of this software for RGB color, single, not stack images, https://imagej.nih.gov/ij/docs/guide/user-guide.pdf, (accessed on 1 April 2023). One image for each organ (brain and placenta) and the case of the different experimental groups were processed, and a total of 341 slides were analyzed. Quantifications were expressed as the number of positive-stained cells/analyzed area. Histological analyses were performed by researchers who were blind with respect to the information about the cases. The blinding of the data was maintained until the analysis was terminated.

### 4.4. Statistical Analysis

Statistical analysis was performed by using SPSS v.22.0 (IBM Corp., Armonk, NY, USA). The quantitative results of immunohistochemical analyses of brain and placental sections were expressed as the median and interquartile range (IQR). The Mann–Whitney U test was used to compare medians between two independent groups, while the Kruskal–Wallis test was used for comparisons of median values among three or more groups.

A value of *p* < 0.05 was considered statistically significant.

## 5. Conclusions

In this study, we found that the immunohistochemical expression of specific OS markers in the brain cortex and placentas was different between fetuses/newborns undergoing intrauterine, intrapartum, and post-partum death.

iNOS showed the highest expression in brain tissue in fetal intrauterine death, followed by post-partum death, while in intrapartum death, the iNOS immunoreactivity was very low.

Regarding IL-6, fetal intrauterine death showed the highest expression in the brain cortex, while intrapartum death and post-partum death showed weak and similar immunoreactivity.

Therefore, those differences in the immunoreactivity of brain tissue to iNOS and IL-6 between fetal intrauterine death (strong reaction to iNOS and IL-6), intrapartum death (low reaction to iNOS and IL-6) and post-partum death (low reaction to iNOS and moderate reaction to IL-6) may represent a first step in identifying the timing of HIE.

For placenta samples, post-partum death placentas showed the highest immunoreactivity to NOX2, which was almost double that compared to fetal intrauterine death and intrapartum death placentas.

Placental tissues of fetal intrauterine death and intrapartum death showed higher expression of iNOS than post-partum death, while the IL-6 expression was higher in fetal intrauterine death than post-partum death. These differences in the placental expression of iNOS and IL-6 between groups almost confirmed the findings observed in the brain cortex. Therefore, such a diagnostic approach may provide useful information for identifying the timing of the lethal injury above HIE. Immunohistochemistry is becoming increasingly important for forensic purposes, especially in identifying the timing of lesions.

## Figures and Tables

**Figure 1 ijms-24-12221-f001:**
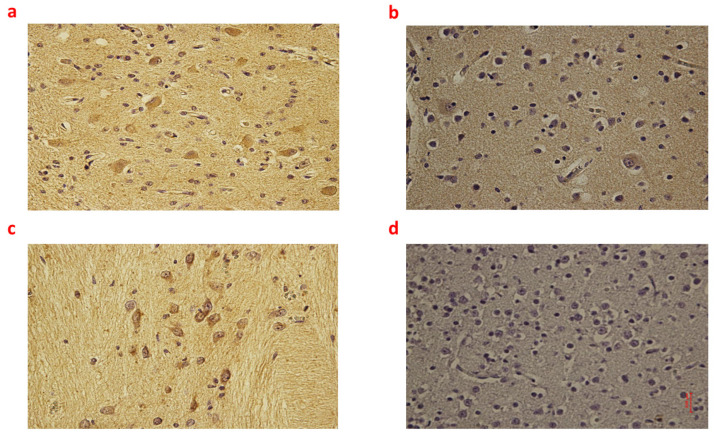
(**a**–**d**): Immunohistochemical expression of inducible Nitric Oxide Synthase (iNOS) in the brain cortex, frontal area: Fetal intrauterine death neuronal and glial expression of iNOS Group_1 B 40× (**a**); Intrapartum death neuronal and glial expression of iNOS Group_2 B 40× (**b**); Post-partum death neuronal expression of iNOS Group_3 B 40× (**c**); Sudden neonatal death negative reaction of iNoS Controls B 40× (**d**).

**Figure 2 ijms-24-12221-f002:**
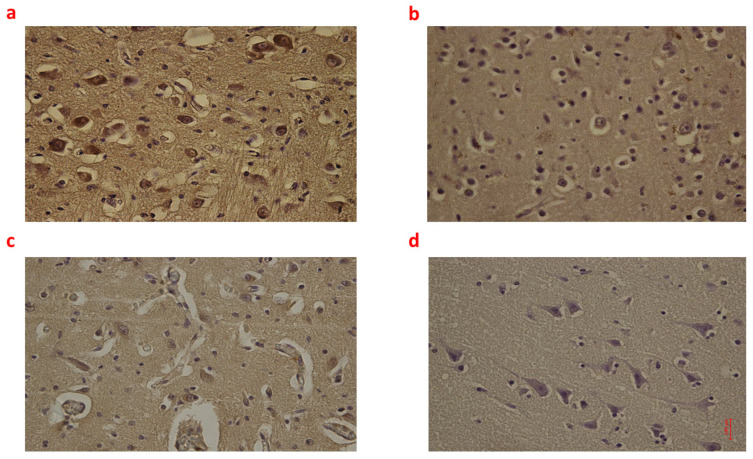
(**a**–**d**): Immunohistochemical expression of Interleukin-6 (IL-6) in the brain cortex, frontal area: Fetal intrauterine death high neuronal and glial expression of IL-6 Group_1 B 40× (**a**); Intrapartum death low neuronal and glial expression of IL-6 Group_2 B 40× (**b**); Post-partum death low neuronal and glial expression of IL-6 Group_3 B 40× (**c**); Sudden neonatal death negative neuronal and glial expression of IL-6 Controls B 40× (**d**).

**Figure 3 ijms-24-12221-f003:**
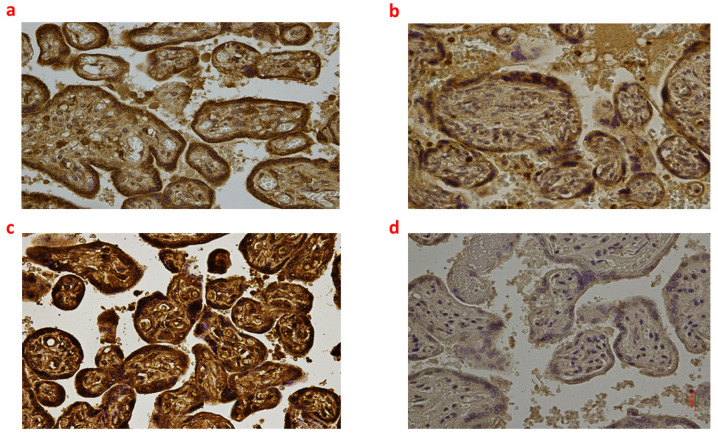
(**a**–**d**): Immunohistochemical expression of NADPH oxidase 2 (NOX2) in the placental tissue, central area: Fetal intrauterine death Group_1 P 40× (**a**); Intrapartum death Group_2 P 40× (**b**); Post-partum death Group_3 P 40× (**c**); Sudden neonatal death Controls P 40× (**d**).

**Figure 4 ijms-24-12221-f004:**
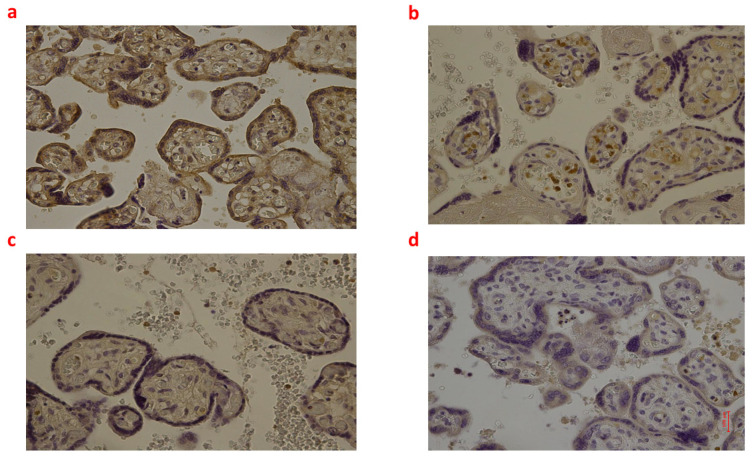
(**a**–**d**): Immunohistochemical expression of inducible Nitric Oxide Synthase (iNOS) in the placental tissue, central area: Fetal intrauterine death Group_1 P 40× (**a**); Intrapartum death Group_2 P 40× (**b**); Post-partum death Group_3 P 40× (**c**); Sudden neonatal death Controls P 40× (**d**).

**Table 1 ijms-24-12221-t001:** Quantitative expression of NADPH oxidase 2 (NOX2), 8-hydroxy-2′-deoxyguanosine (8-OHdG), nitrotyrosine (NT), nitric oxide synthase (iNOS), and IL-6 in the brain cortex among groups. Data reported as median with interquartile range (IQR) of the percentage of the number of positive colored cells/microscopic area analyzed. *p* values are calculated by the Kruskal–Wallis test. Comparisons between two independent groups were made by the Mann–Whitney U test.

	Group_1 B (n = 9)	Group_2 B (n = 8)	Group_3 B (n = 6)	Controls B (n = 6)	*p*-Value
NOX2	4.549 (7.785) ^1^	4.303 (7.132) ^1^	6.841 (2.683) ^1^	0.258 (1.307)	0.009
8-OHDG	11.130 (4.294) ^1^	6.410 (3.976) ^1^	10.800 (7.906) ^1^	1.040 (1.065)	0.003
NT	6.926 (5.175) ^1^	4.239 (3.390) ^1^	2.729 (3.292) ^1^	0.140 (0.062)	0.005
iNOS	3.851 (2.697) ^1,2,3^	1.037 (1.424) ^2,4^	2.240 (1.146) ^1,3,4^	0.723 (0.643)	<0.001
IL-6	4.501 (6.486) ^1,2,3^	0.664 (0.518) ^1,2^	0.535 (0.253) ^1,3^	0.100 (0.046)	0.001

^1^ Group_1 B, Group_2 B, Group_3 B vs. Controls B: Statistically significant. ^2^ Group_1 B vs. Group_2 B: Statistically significant. ^3^ Group_1 B vs. Group_3 B: Statistically significant. ^4^ Group_2 B vs. Group_3 B: Statistically significant.

**Table 2 ijms-24-12221-t002:** Quantitative expression of NADPH oxidase 2 (NOX2), 8-hydroxy-2′-deoxyguanosine (8-OHdG), nitrotyrosine (NT), nitric oxide synthase (iNOS), and IL-6 in the placental tissue among groups. Data reported as median with interquartile range (IQR) of the percentage of the number of positive colored cells/microscopic area analyzed. *p* values are calculated by the Kruskal–Wallis test. Comparisons between two independent groups were made by the Mann–Whitney U test.

	Group_1 P (n = 9)	Group_2 P (n = 8)	Group_3 P (n = 6)	Controls P (n = 6)	*p*-Value
NOX2	10.875 (9.920) ^3^	11.839 (6.369)	20.519 (9.974) ^1,3,4^	8.588 (3.572)	0.01
8-OHDG	4.107 (11.105)	6.151 (5.658)	6.504 (2.657) ^1^	2.861 (3.138)	0.201
NT	8.349 (14.111)	13.943 (12.504) ^1^	15.979 (3.661) ^1^	4.882 (3.560)	0.019
iNOS	1.714 (2.199) ^1,3^	0.714 (0.850) ^1,4^	0.154 (0.146) ^3,4^	0.105 (0.087)	<0.001
IL-6	0.168 (3.355) ^3^	0.004 (0.876)	0.020 (0.028) ^3^	0.057 (0.192)	0.026

^1^ Group_1 P, Group_2 P, Group_3 P vs. Controls: Statistically significant. ^2^ Group_1 P vs. Group_2 P: Statistically significant. ^3^ Group_1 P vs. Group_3 P: Statistically significant. ^4^ Group_2 P vs. Group_3 P: Statistically significant.

**Table 3 ijms-24-12221-t003:** General characteristics of included subjects. Cause of death column lists the disease/injury considered as responsible for death at forensic investigation. The Group of each case is indicated in the table.

Patient Id	Group	Gender	Gestational Age at Birth (Weeks + Days)	Exitus (Type)	Birth (Type)	Weight (Grams)	Survival Time	Apgar Score	Cause of Death
1	1	Female	40 + 2	Intrauterine	Cesarean	2749	-	-	Placental infarction
2	1	Female	38	Intrauterine	Vaginal	3300	-	-	Fetal asphyxia
3	1	Male	39 + 2	Intrauterine	Vaginal	2680	-	-	Fetal asphyxia
4	1	Female	38	Intrauterine	Cesarean	3000	-	-	Placental infarction
5	1	Female	38	Intrauterine	Vaginal	3359	-	-	Heart failure
6	1	Female	39 + 4	Intrauterine	Vaginal	3200	-	-	Fetal asphyxia
7	1	Male	39 + 1	Intrauterine	Cesarean	4045	-	-	Fetal asphyxia
8	1	Female	40	Intrauterine	Vaginal	2996	-	-	Fetal asphyxia
9	1	Male	37 + 6	Intrauterine	Vaginal	3680	-	-	Fetal asphyxia
10	2	Male	41 + 3	Intrapartum	Cesarean	3129	-	-	Prolonged umbilical cord compression
11	2	Female	41 + 2	Intrapartum	Cesarean	3850	-	-	Unexplained fetal death
12	2	Female	41	Intrapartum	Cesarean	3800	40 min	-	Acute hyschemia
13	2	Female	38	Intrapartum	Vacuum-assisted vaginal	4710	-	1′: 0 5′: 0	Acute asphyxia due to umbilical cord knot
14	2	Female	36	Intrapartum	Vaginal	2500	-	-	Acute asphyxia due to umbilical cord loop
15	2	Female	41 + 3	Intrapartum	Cesarean	3050	-	-	Acute asphyxia due to umbilical cord loop
16	2	Female	41	Intrapartum	Cesarean	2800	-	-	Acute asphyxia due to umbilical cord compression
17	2	Male	41	Intrapartum	Cesarean	3450	-	1′: 0	Brain edema
18	2	Female	41	Intrapartum	Cesarean	2800	-	-	Acute asphyxia due to umbilical cord compression
19	3	Female	40 + 1	Post-partum	Vaginal	3320	6 h	1′: 6 5′: 8	Acute asphyxia due to umbilical cord compression
20	3	Female	41 + 3	Post-partum	Vaginal	3000	6 h	1′: 2 5′: 1	Pneumonia
21	3	Male	38 + 4	Post-partum	Vaginal	3390	3 days	1′: 0 5′: 0 10′: 2 20′: 4	Cerebral hemorrhage
22	3	Male	36 + 2	Post-partum	Vacuum-assisted vaginal	2400	1 day	1′: 9 5′: 10	Respiratory distress
23	3	Female	38 + 4	Post-partum	Cesarean	2900	2 days	1′: 1 5′: 1	Cerebral hemorrhage
24	3	Female	41 + 6	Post-partum	Cesarean	3600	1 day	1′: 2 5′: 3	Meconium aspiration syndrome
25	3	Female	40 + 1	Post-partum	Vaginal	3320	6 h	1′: 6 5′: 8	Respiratory distress
26	Controls	Male	37	Suicide (maternal hanging)	-	2750	-	-	Placental insufficiency
27	Controls	Female	38	Maternal thromboembolism	-	2900	-	-	Acute hypoxia
28	Controls	Male	38 + 3	Cardiac malformation	Cesarean	2835	-	-	Aortic coarctation
29	Controls	Female	36 + 1	Maternal aortic aneurysm rupture	-	2615	-	-	Placental insufficiency
30	Controls	Female	41 + 3	Maternal thromboembolism	-	3512	-	-	Acute hypoxia
31	Controls	Female	38 + 2	Cardiac malformation	Cesarean	2904	-	-	Aortic coarctation

**Table 4 ijms-24-12221-t004:** Antibody dilution.

Antibody against	Concentration of Primary Antibody
NOX2 (Santa Cruz, CA, USA)	1:50
8-OHdG (JaICA, Tokyo, Japan)	1:10
NT (Santa Cruz, CA, USA)	1:600
iNOS (Santa Cruz, CA, USA)	1:100
IL-6 (Santa Cruz, CA, USA)	1.200
AQP4 (Santa Cruz, CA, USA)	1:200

## Data Availability

Not applicable.

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
