# Peer review of "Oxidative Stress Markers in Human Brain and Placenta May Reveal the Timing of Hypoxic-Ischemic Injury: Evidence from an Immunohistochemical Study"

_ijms, 2023, doi:10.3390/ijms241512221_

Round 1
Reviewer 1 Report
Comments: Oxidative stress markers in human brain and placenta may reveal the timing of hypoxic-ischemic injury: Evidence from an immunohistochemical study
At first, I appreciate the objective of this study, but there are a lot of flaws and a lack of coherency in the whole write-up, which causes a loss of interest for the reader. It gives the impression that it is being written very carelessly.
- The abstract is poorly written. No coherence among sentences. The author mentioned group -1 and GROUP_3 but did not mention what they are supposed to be. At least the group’s specifications should be mentioned in the abstract clearly before describing the findings.
- The background is not powerful enough to convince the reason for conducting this study. it is stated that the purpose is to identify the timing of hypoxic-ischemic insult underlying HIE to avoid medical liability. What kind of medical liability is in focus?
- This study is supposed to find the timing of hypoxic-ischemic injury. But, you already know about fetal/neonatal death whether its intrauterine, prepartum or postpartum. That is definitely known to pathologists because dead subject data is available to them too. Then, why is the need for this study to conduct?
- The significance of the study is not clearly mentioned.
- In the Result section, Table 1 is completely unfit here. It should be part of the Methods because it presents the features and characteristics of study groups.
- AQP4 is a well-known astrocyte marker. I wonder, What is the purpose of considering this marker in this study? the focus of the study is to detect Oxidative stress markers. I could not understand the significance of using this marker.
- In Tables 2 and 3, different groups are presented by codes i.e., Group_1. In Figures 1 and 2, groups are presented with their exitous type names. That is very confusing. It gives a feeling of, different people write at different times and before submission nobody bothers to check clearance in the writ-up. The author should use the same kind of style in the whole manuscript.
- In all figures, micrographs’ specifications are missing everywhere (magnification? Area? Etc). images appear with poor resolution. Not as much good as they should be to be acceptable for publication.
- Graphical presentation of results makes it easy to understand and compare the finding among groups but they are added as supplemental figures.
- In the Materials and Method section, spelling mistake in line 266; I think “then in cubed g the primary” should be replaced with “then incubated with primary”.
- Line 277, sentences ended with two full stops.
- Line 267-272 lines are so complicated. Rewrite to clear the method. Also mention company name for the kit used in the study.
- The discussion is a little bit better, but it needs improvement. for instance, highlight the significance of using AQP4. Instead of using group IDs in discussion mention their specificity e.g., intrauterine, prepartum so that it could be more understandable to compare the findings.
- The conclusion is very ambiguous and unclear.
The author should conclude the study by nominating markers that may going to be significantly expressed in the corresponding group. Then it may be helpful for pathologists and justify the purpose of this study. otherwise, nothing is clearly justifiable in the current scenario with current findings.
English quality is ok, but the flow of information and way of presentation is unclear. There is a lack of coherence in manuscript.
Author Response
Point-by-point reply to the Reviewers’ comments
We appreciate the Reviewers’ comments and found them very helpful to further improve our manuscript. We revised our manuscript accordingly. Changes were fully incorporated in the appropriate sections of the revised manuscript and highlighted as “Track changes”. A point by point reply to the Reviewers’ comments is provided below.
Reviewer1 (R1):
Comments: Oxidative stress markers in human brain and placenta may reveal the timing of hypoxic-ischemic injury: Evidence from an immunohistochemical study
At first, I appreciate the objective of this study, but there are a lot of flaws and a lack of coherency in the whole write-up, which causes a loss of interest for the reader. It gives the impression that it is being written very carelessly.
The authors would like to thank the anonymous reviewers for their suggestions and comments aimed at improving the manuscript. Herewith are our answers to the specific comments:
-The abstract is poorly written. No coherence among sentences. The author mentioned group -1 and GROUP_3 but did not mention what they are supposed to be. At least the group’s specifications should be mentioned in the abstract clearly before describing the findings.
We totally modified the abstract see in the manuscript the yellow highlighted text.
-The background is not powerful enough to convince the reason for conducting this study. it is stated that the purpose is to identify the timing of hypoxic-ischemic insult underlying HIE to avoid medical liability. What kind of medical liability is in focus?
-This study is supposed to find the timing of hypoxic-ischemic injury. But, you already know about fetal/neonatal death whether its intrauterine, prepartum or postpartum. That is definitely known to pathologists because dead subject data is available to them too. Then, why is the need for this study to conduct?
Your two questions are very important and helpful, we want to individuate markers useful to support the clinical data in cases of medical malpractice in the obstetric field. For better clarity of the reason for our study, it is important to explain the role of forensic pathologists in Italy. Italian law provides that medical liability is assessed according to the Penal and Civil Codes. In Italy, the obstetric physician frequently can be investigated by the prosecutor's office in case of fetal or newborn death during labor or in the last part of pregnancy during medical activity. So, a doctor could be convicted of manslaughter or negligent homicide, and the evaluation must be more correct and scientifically supported possible. The prosecutor and the judge appoint a forensic pathologist and an obstetrician, very qualified, to perform the autopsy on the fetus or newborn and study deeply the clinical documentation for the evaluation of the conduct of the doctors who carried out the birth and followed the pregnancy. The evaluation must be scrupulous and supported by the literature, so we think it is crucial to provide further scientific evidence for the evaluation of the cases, using immunohistochemical markers. The second point is the civil law consequence of a penal law case because the parents very frequently claim compensation for the damage suffered from the death or serious disability of the newborn. The compensation for damage in the obstetric field is the highest among the cases of medical malpractice and involves the compensation of millions of euros. For all the reasons illustrated both for Penal Law in Italy and for Civil law or compensatory damages in other countries this study could be very useful for the support of cases of medical malpractice in obstetrics.
-The significance of the study is not clearly mentioned.
Thank you again for your previous questions, we understood that the paper was not clear on this point, and we have integrated the manuscript to better explain the significance of our study, see the yellow highlighted text in the introduction and discussion.
-In the Result section, Table 1 is completely unfit here. It should be part of the Methods because it presents the features and characteristics of study groups.
We have transferred Table 1 from the Results to the Methods section. Tables are renumbered, now Table 1 is Table 2, see the yellow highlighted text in the manuscript.
-AQP4 is a well-known astrocyte marker. I wonder, What is the purpose of considering this marker in this study? the focus of the study is to detect Oxidative stress markers. I could not understand the significance of using this marker.
Some authors performed previous studies and published data on the role of AQPS in forensic pathology, we see in vitro studies about the edema that occurs during the early stage of ischemic stroke. The AQP4 plays an important role in brain edema, so in a preliminary experiment, we tested AQP4 in our newborn and fetus brain samples. The immunohistochemical of AQP4 was not significant because the reaction in the three Groups of cases and in the Control Group was negative, so the AQP4 was discarded as a possible marker and focalized our study on oxidative stress markers. For completeness and scientific correctness, it seemed right to point out the negative data of the AQP4.
-In Tables 2 and 3, different groups are presented by codes i.e., Group_1. In Figures 1 and 2, groups are presented with their exitous type names. That is very confusing. It gives a feeling of different people write at different times and before submission nobody bothers to check clearance in the writ-up. The author should use the same kind of style in the whole manuscript.
We clarify the groups in the text and modify the manuscript to insert a unique code for each group for brain e placenta samples, see the yellow highlighted text in the manuscript.
-In all figures, micrographs’ specifications are missing everywhere (magnification? Area? Etc). images appear with poor resolution. Not as much good as they should be to be acceptable for publication.
We insert the magnification and the area, see the yellow highlighted text in the manuscript. The poor resolution of images we think is related to the web version of the manuscript with figures embedded in the text in Word version. We prepared the figures in high resolution and insert the scale bar, according to the suggestion of another review, and change the figures with ones with better resolution and with a scale bar.
-Graphical presentation of results makes it easy to understand and compare the finding among groups but they are added as supplemental figures.
We are agreeing with you, but we must be a choice because the paper is too long and we inserted the graphical presentation of results in supplemental figures.
-In the Materials and Method section, spelling mistake in line 266; I think “then in cubed g the primary” should be replaced with “then incubated with primary”.
We correct the text as you suggest, see the yellow highlight text, in the current manuscript version is line 330.
-Line 277, sentences ended with two full stops.
We correct the text as you suggest, see the yellow highlight text, in the current manuscript version is line 340.
-Line 267-272 lines are so complicated. Rewrite to clear the method. Also mention company name for the kit used in the study.
We rewrote the text as you suggest, see the yellow highlight text, in the current manuscript version are lines 330-335.
-The discussion is a little bit better, but it needs improvement. for instance, highlight the significance of using AQP4. Instead of using group IDs in discussion mention their specificity e.g., intrauterine, prepartum so that it could be more understandable to compare the findings.
We improve the discussion and clarify the ID of the groups to make the text easier to read and intelligible, see the yellow highlighted text in the manuscript.
-The conclusion is very ambiguous and unclear.
We rewrote the conclusion as you suggest, to make the text easier to read and intelligible, see the yellow highlight text.
The author should conclude the study by nominating markers that may going to be significantly expressed in the corresponding group. Then it may be helpful for pathologists and justify the purpose of this study. otherwise, nothing is clearly justifiable in the current scenario with current findings.
See the conclusion we improve the text, as you suggest, to make the text easier to read and intelligible, see the yellow highlight text.
We hope the improvement of the manuscript performed using your very good suggestion rendered the text clearer and complete, and the manuscript better.

Reviewer 2 Report
The manuscript of Baldari et al. provides an overview of a retrospective study conducted on fetal/neonatal brains and placental samples to investigate the role of oxidative stress in the pathogenesis and progression of hypoxic ischemic encephalopathy (HIE). This study highlights the challenge faced by forensic pathologists in identifying the timing of hypoxic-ischemic insult leading to HIE, and emphasizes the importance of this information for medical liability. The study employed immunohistochemical investigations for various markers, including AQP4, NOX2, 8-OHdG, NT, iNOS, and IL-6, to assess their expression in different groups.
In the present form, the manuscript is difficult to understand. I listed all my comments and the order in which they are presented is not related to their importance.
I will comment on the abstract with a few suggestions for improvement of the abstract, but it would also apply to some parts of the main manuscript:
Clarity and conciseness: The abstract would benefit from clearer and more concise language to enhance its readability. Consider rephrasing complex sentences (for instance sentence lane 36-37), or naïve sentences (“AQP4 was not helpful” [lane 35]), or meaningless sentences (“… almost confirmed our findings observed in the brain cortex” [lane 38-39]).
In addition to specify the study design It would be helpful to explicitly mention some details regarding the groups, sample size, selection criteria, and study population.
Provide context for markers: When mentioning the markers investigated in the study (AQP4, NOX2, 8-OHdG, NT, iNOS, and IL-6), it would be useful to provide a brief explanation of their relevance and their association with oxidative stress and HIE.
Beyond the abstract, clarity and conciseness should be also applied. A clearer sentence is required in lane 90 of the results section, which could be changed to “Semi-quantitative assessment for aquaporin-4 (AQP4) through immunohistochemical analyses on brain cortex samples…”.
And in the following sentence, what does mean “… gave negative results…” (lane 91)?
Why the analysis of the expression for NOX2, 8-OHDG, NT, iNOS and IL-6 is quantitative and for AQP4 is semiquantitative?
Please in results section when presenting table 1, could the authors define case-Group 1, 2 and 3, and controls? I guess they are based on the type of Exitus, but could the authors please write just a sentence to make it clear. I am aware of its definition in the Material and Methods section, but as a reader I need this little piece of information as I read the manuscript. I think authors must start their results section explaining a little bit the population of study and the groups.
In tables 2 and 3, since the authors used the Kruskal-Wallis test, they have to perform multiple comparison of medians between groups with post-hoc analysis instead of Mann-Whitney U test. Please, reanalyse the data with correct statistic tests. The information from statistical tests should be included in the supplementary figures, too.
Legend of table 2 and 3 must indicate the units of measured values, and that values refer to median (IQR).
Did images in figure 1 a, b, c, d match with group 1, 2 ,3 and controls, respectively? If yes, the Figure 1d (control) shows the same levels of iNOS (or even higher levels) as the Figure 1a, b, c (case-groups). Thus, based on the immunohistochemistry images, no differences of iNOS expression in brain cortex were observed between the case-groups and controls.
For figures 1 to 4, authors must indicate that a, b, c and d images are related to case-groups and controls.
How many fields did the authors count per sample? Number of counted fields should be indicated in methods section. It would be preferable to have normalised data showing the number of HRP-DAB iNOS positive cells per total number of cells (hematoxylin staining).
Please, authors must show the scale bar in the microphotographs of figures 1 to 4. It seems that image on Figure 2d has greater magnification.
When mentioning the oxidative stress markers investigated in the study (AQP4, NOX2, 8-OHdG, NT, iNOS, and IL-6), it would be useful to provide an explanation of their use in the introduction.
Results and findings: The data shows the expression levels of 5 oxidative markers in different groups of samples that corresponded to different types of foetal deaths. Most of the markers are expressed in the three groups of cases without any association with a precise timing of hypoxic-ischemic injury. Authors does not provide any conclusion drawn from the stress oxidative markers and the timing of expression. It would be valuable to include key results or significant observations about their specific relevance and their association with HIE.
The results show the data redundantly. The same data are presented in table 2, Figures 1 and 2 and in supplementary figures S1a-S5a for brain cortex; and table 3, Figures 3-4, and supplementary figures S1b-S5b for placenta. Missing information or comments from the immunoshistochemical images, and cells that express these markers.
More importantly, by looking at the immunohistochemical images in detail, no differences in the expression level of oxidative stress markers could be observed between the three case-groups and controls. For example, Images of iNOS expression in the placenta (Figure 4) are not representative of the quantification shown in Figure S4b. Group 1 (intrauterine dead) is presented as having the highest expression of iNOS (Figure S4b); however, this result does not match what can be observed in the images in Figure 4.
Implications and significance: Because the timing of hypoxic-ischemic insult was unknown, how did the authors associate presence of oxidative markers with the occurrence of this insult?
Section 4.3 Methods lane 266 presents typographical errors and lane 267 must read table 4.
Because the authors choose to use immunohistochemistry to quantify oxidative stress markers, a more detailed description of the method is required for reproducibility. Are all tissue sections stained for a specific marker performed at the same time? Why do immnunoshistochemical images have a different background colour? Does the quantification with Image J take this different background colour into account? A better description of methodology is required.
Author Response
Reviewer2 (R2): The manuscript of Baldari et al. provides an overview of a retrospective study conducted on fetal/neonatal brains and placental samples to investigate the role of oxidative stress in the pathogenesis and progression of hypoxic ischemic encephalopathy (HIE). This study highlights the challenge faced by forensic pathologists in identifying the timing of hypoxic-ischemic insult leading to HIE, and emphasizes the importance of this information for medical liability. The study employed immunohistochemical investigations for various markers, including AQP4, NOX2, 8-OHdG, NT, iNOS, and IL-6, to assess their expression in different groups.
In the present form, the manuscript is difficult to understand. I listed all my comments and the order in which they are presented is not related to their importance.
The authors would like to thank the anonymous reviewer for their suggestions and comments aimed at improving the manuscript. Herewith are our answers to the specific comments:
We perform an improvement of the manuscript, we hope to render the text clearer and complete, and the manuscript better.
I will comment on the abstract with a few suggestions for improvement of the abstract, but it would also apply to some parts of the main manuscript:
Clarity and conciseness: The abstract would benefit from clearer and more concise language to enhance its readability. Consider rephrasing complex sentences (for instance sentence lane 36-37), or naïve sentences (“AQP4 was not helpful” [lane 35]), or meaningless sentences (“… almost confirmed our findings observed in the brain cortex” [lane 38-39]).
In addition to specify the study design It would be helpful to explicitly mention some details regarding the groups, sample size, selection criteria, and study population.
Provide context for markers: When mentioning the markers investigated in the study (AQP4, NOX2, 8-OHdG, NT, iNOS, and IL-6), it would be useful to provide a brief explanation of their relevance and their association with oxidative stress and HIE.
Beyond the abstract, clarity and conciseness should be also applied. A clearer sentence is required in lane 90 of the results section, which could be changed to “Semi-quantitative assessment for aquaporin-4 (AQP4) through immunohistochemical analyses on brain cortex samples…”.
And in the following sentence, what does mean “… gave negative results…” (lane 91)?
We totally modified the abstract according to your very good suggestions, see in the manuscript the yellow highlighted text.
Why the analysis of the expression for NOX2, 8-OHDG, NT, iNOS and IL-6 is quantitative and for AQP4 is semiquantitative?
Thank you for your suggestion, we performed a preliminary semiquantitative analysis for all markers to eliminate non-significant results, and after we perform the quantitative analysis. We insert this information in the text to improve the manuscript, See the yellow highlighted text in the manuscript.
Please in results section when presenting table 1, could the authors define case-Group 1, 2 and 3, and controls? I guess they are based on the type of Exitus, but could the authors please write just a sentence to make it clear. I am aware of its definition in the Material and Methods section, but as a reader I need this little piece of information as I read the manuscript. I think authors must start their results section explaining a little bit the population of study and the groups.
According to the suggestion of another review, we change the position of Table 1 from the Results to the Methods section. Tables are renumbered, now table 1 is table 2. We define the case Group 1, 2 and 3 in the abstract and repeat in the results, see the yellow highlighted text in the manuscript.
In tables 2 and 3, since the authors used the Kruskal-Wallis test, they have to perform multiple comparison of medians between groups with post-hoc analysis instead of Mann-Whitney U test. Please, reanalyse the data with correct statistic tests. The information from statistical tests should be included in the supplementary figures, too.
Legend of table 2 and 3 must indicate the units of measured values, and that values refer to median (IQR).
Thank you for the opportunity to improve the manuscript with your suggestions. We have added the measure of central tendency and the units of measurement in Table 2 and Table 3, as requested. Regarding the statistical analysis, it was conducted by a statistics professor, and no methodological errors were found. The post-hoc analysis using the Mann-Whitney U test is considered equally valid compared to the Tukey's range test. The rationale for using the Mann-Whitney U test as a post-hoc test after demonstrating statistical significance with the Kruskal-Wallis test stems from the need to perform pairwise comparisons between groups to determine specific differences.
The Kruskal-Wallis test is a non-parametric test used to analyze differences among three or more independent groups when the dependent variable is not normally distributed. It tests the null hypothesis that the populations have equal medians. If the Kruskal-Wallis test shows a statistically significant result, it suggests that there are differences between the groups. To further investigate which specific groups differ from each other, post-hoc tests are performed. The Mann-Whitney U test, also known as the Wilcoxon rank-sum test, is a suitable post-hoc test for pairwise comparisons between two independent groups. It assesses whether the distributions of the two groups differ significantly. By applying the Mann-Whitney U test to pairs of groups, we can determine which groups have significant differences in their medians. Here are two scientific references that support the rationale for using the Mann-Whitney U test as a post-hoc test after obtaining statistical significance with the Kruskal-Wallis test: 1) Conover, W. J., & Iman, R. L. (1981). Rank transformations as a bridge between parametric and nonparametric statistics. The American Statistician, 35(3), 124-129. doi: 10.1080/00031305.1981.10479327; 2) Zar, J. H. (2010). Biostatistical analysis (5th ed.). Prentice Hall.
These references discuss the application of non-parametric tests, including the Kruskal-Wallis test and the Mann-Whitney U test, and their use in analyzing differences between groups. In summary, after obtaining statistical significance with the Kruskal-Wallis test, the Mann-Whitney U test is employed as a post-hoc test to determine specific pairwise differences between groups. The cited references provide further information on the use of these tests in analyzing non-parametric data.
Did images in figure 1 a, b, c, d match with group 1, 2 ,3 and controls, respectively? If yes, the Figure 1d (control) shows the same levels of iNOS (or even higher levels) as the Figure 1a, b, c (case-groups). Thus, based on the immunohistochemistry images, no differences of iNOS expression in brain cortex were observed between the case-groups and controls.
For figures 1 to 4, authors must indicate that a, b, c and d images are related to case-groups and controls.
The figure 1 show the group and the controls, we add the groups in the legend. For better accuracy, we performed a semiquantitative analysis and then a quantitative analysis the figure represents results of our study: for iNOS, the highest expression in brain tissue was observed in Fetal intrauterine death (Group_1 B) followed by Post-partum death (Group_3 B), while in Intra-partum death (Group_2 B) the iNOS immunoreactivity was very low.
How many fields did the authors count per sample? Number of counted fields should be indicated in methods section. It would be preferable to have normalised data showing the number of HRP-DAB iNOS positive cells per total number of cells (hematoxylin staining).
We insert in the text the data about the quantitative analysis that you ask us, see in the manuscript the yellow highlighted text.
Please, authors must show the scale bar in the microphotographs of figures 1 to 4. It seems that image on Figure 2d has greater magnification.
We insert the scale bar in the figures and the magnification in the legend, all field was acquired with the magnification 40x.
When mentioning the oxidative stress markers investigated in the study (AQP4, NOX2, 8-OHdG, NT, iNOS, and IL-6), it would be useful to provide an explanation of their use in the introduction.
We integrated the introduction with the markers
Results and findings: The data shows the expression levels of 5 oxidative markers in different groups of samples that corresponded to different types of foetal deaths. Most of the markers are expressed in the three groups of cases without any association with a precise timing of hypoxic-ischemic injury. Authors does not provide any conclusion drawn from the stress oxidative markers and the timing of expression. It would be valuable to include key results or significant observations about their specific relevance and their association with HIE.
We improved the manuscript with the information that you suggest in particular the conclusion, see in the manuscript the yellow highlighted text.
The results show the data redundantly. The same data are presented in table 2, Figures 1 and 2 and in supplementary figures S1a-S5a for brain cortex; and table 3, Figures 3-4, and supplementary figures S1b-S5b for placenta. Missing information or comments from the immunoshistochemical images, and cells that express these markers.
Thanks for your suggestion, we insert the information in the legends of figure 1 and 2, see in the manuscript the yellow highlighted text.
More importantly, by looking at the immunohistochemical images in detail, no differences in the expression level of oxidative stress markers could be observed between the three case-groups and controls. For example, Images of iNOS expression in the placenta (Figure 4) are not representative of the quantification shown in Figure S4b. Group 1 (intrauterine dead) is presented as having the highest expression of iNOS (Figure S4b); however, this result does not match what can be observed in the images in Figure 4.
Thank you very much for this comment and for your careful evaluation of our manuscript which prompted us to replace, in the revised version of the manuscript, the image included in panel A of Fig.4 of the original version with a more representative one, which is visually more the highest expression of iNOS.
Implications and significance: Because the timing of hypoxic-ischemic insult was unknown, how did the authors associate presence of oxidative markers with the occurrence of this insult?
Thank you, your question is the central point of our paper, as you can see in table 3 (table 1 in the previous version of the paper) we have the clinical data of all cases and we can perform a correlation between the time of hypoxic insult and the time of the markers. We hope, that after reading the revised version of the paper the text is more clear and complete.
Section 4.3 Methods lane 266 presents typographical errors and lane 267 must read table 4.
We correct the errors in the text, see the yellow highlight text, in the current manuscript version is lines 330 and 331.
Because the authors choose to use immunohistochemistry to quantify oxidative stress markers, a more detailed description of the method is required for reproducibility. Are all tissue sections stained for a specific marker performed at the same time? Why do immnunoshistochemical images have a different background colour? Does the quantification with Image J take this different background colour into account? A better description of methodology is required.
We improve the method section with the suggestion that you require, see the yellow highlight text in the manuscript.
We performed the analysis in more days because the slides were 62 (31 brains and 31 placenta) for each marker except for AQP4 (only brain). We performed one marker for each organ for session for a total 11 sessions. The reagents, the laboratory, and the operators were the same for all sessions. We see the difference in background and it is very frequent when you work with autoptic material, in particular in newborn or fetus cases. This is normal in forensic pathology samples and we take this different background colour into account.

Round 2
Reviewer 1 Report
In the current situation, the manuscript looks ok. In paragraph (96-101), I found sentence-making issues that is making hard to understand what the author want to say, because of missing and inappropriate punctuation. and a spelling mistake like "Aquaporine a". I guess It should be Aquaporin 4.
Looks ok.
Author Response
Point-by-point reply to the Reviewers’ comments
We appreciate the Reviewers’ second-round comments and found them very helpful in further improving our manuscript. We revised our manuscript accordingly. Changes were fully incorporated in the appropriate sections of the revised manuscript and highlighted as “Track changes” in a different color, the revisions of the first round were highlighted in yellow, while the second-round revisions are highlighted in green. A point-by-point reply to the Reviewers’ comments is provided below.
Reviewer1 (R1):
Comments and Suggestions for Authors: In the current situation, the manuscript looks ok. In paragraph (96-101), I found sentence-making issues that is making hard to understand what the author want to say, because of missing and inappropriate punctuation. and a spelling mistake like "Aquaporine a". I guess It should be Aquaporin 4.
Comments on the Quality of English Language: Looks ok.
We really appreciate that your suggested revisions to the paper have made the manuscript better.
We thank you for your further revisions, we perform the changes in the manuscript, you can see the second round track changes highlighted in green in the text (see lines 96-100).

Reviewer 2 Report
The authors have tried to modify the manuscript to improve its poor readability. As a reviewer I am more concerned with the scientific data. I think it is up to the 15 co-authors, and perhaps the guest-editors, to do their best to better communicate their results, primarily the abstract. I tried to help the authors on this issue in my first review, but I feel like I failed.
The authors did not address my scientific concerns about whether the quantification is done correctly. How the background was normalized between images? Looking at the images in the manuscript, it could be not stated that oxidative markers are expressed more in case-groups than in controls. The authors even confirmed that they changed some images from first version to be clearer. Relevant information about blinded experiments is now mentioned, but not in the first version. The authors did not respond to my request to count the number of positive cells per total number of cells. The authors did not answer the number of fields or the number of images per marker they analysed. All images must show the original scale bar.
The authors performed statistical analysis using Kruskal-Wallis test to analyse differences between the four groups. When using this test, it is as easy to analyze differences between groups by making multiple comparison of medians between groups with post-hoc analyses. The authors preferred to waste more time explaining the basics of statistics to me than actually doing the requested test. And they did not incorporate this statistical information in the supplementary figures. This information is key to understanding, very quickly, where the differences lie.
All pages of the manuscript contain text highlighted in yellow. The authors should indicate more specifically what changes they have made to the requested questions.
Author Response
Reviewer2 (R2):
Comments and Suggestions for Authors: The authors have tried to modify the manuscript to improve its poor readability. As a reviewer I am more concerned with the scientific data. I think it is up to the 15 co-authors, and perhaps the guest-editors, to do their best to better communicate their results, primarily the abstract. I tried to help the authors on this issue in my first review, but I feel like I failed.
We really appreciate the very good suggestions performed from you and the other review, we rewrite the abstract and a big part of the manuscript according to your indication, you can see the changes were incorporated in the revised manuscript and highlighted in yellow in the first round of revisions and green in the second round of revisions, as “Track changes”.
The authors did not address my scientific concerns about whether the quantification is done correctly. How the background was normalized between images? Looking at the images in the manuscript, it could be not stated that oxidative markers are expressed more in case-groups than in controls. The authors even confirmed that they changed some images from first version to be clearer. Relevant information about blinded experiments is now mentioned, but not in the first version. The authors did not respond to my request to count the number of positive cells per total number of cells. The authors did not answer the number of fields or the number of images per marker they analysed. All images must show the original scale bar.
Thank you very much for this comment. The required information has been added to the revised version of the manuscript, section Materials and Methods, paragraph “4.3. Methods” (please, see lines 3339-347 “Quantification of NOX2, 8OHdG, NT, iNOS and IL-6 positive-stained cells was performed by the ImageJ software (imagej.nih.gov/ij/), as previously described [53,54], using the “Manual Cell Counting and Marking” protocol of this software for RGB color, single, not stack images (https://imagej.nih.gov/ij/docs/guide/user-guide.pdf). One image for each organ (brain and placenta) and case of the different experimental groups was processed. Quantifications were expressed as number of positive-stained cells/analyzed area. Histological analyses were performed by researchers who were blind with respect to the information about the cases. The blinding of the data was maintained until the analysis was terminated.”)
Getting into the specifics the background of each figure is the original, the normalization was performed using the software ImageJ, the method is the same and the markers analyzed are the same as other published papers (Schiavone S., Neri M., Trabace L., Turillazzi E. The NADPH oxidase NOX2 mediates loss of parvalbumin interneurons in traumatic brain injury: Human autoptic immunohistochemical evidence. Sci. Rep. 2017;7:8752. doi: 10.1038/s41598-017-09202-4; Schiavone S, Neri M, Maffione AB, Frisoni P, Morgese MG, Trabace L, Turillazzi E. Increased iNOS and Nitrosative Stress in Dopaminergic Neurons of MDMA-Exposed Rats. Int J Mol Sci. 2019 Mar 12;20(5):1242. doi: 10.3390/ijms20051242.)
Quantification of positive-stained cells was performed by the ImageJ software, one image for each organ (brain and placenta) and case of the different experimental groups was processed a total of 341 slides was analyzed. We insert the scale bar in each of the four histologic figures (see image d of figure 1,2,3,4).
The authors performed statistical analysis using Kruskal-Wallis test to analyse differences between the four groups. When using this test, it is as easy to analyze differences between groups by making multiple comparison of medians between groups with post-hoc analyses. The authors preferred to waste more time explaining the basics of statistics to me than actually doing the requested test. And they did not incorporate this statistical information in the supplementary figures. This information is key to understanding, very quickly, where the differences lie.
We sincerely thank the editor for his valuable suggestion. The information requested was now added to the captions of the supplementary figures (see the text highlighted in yellow).
All pages of the manuscript contain text highlighted in yellow. The authors should indicate more specifically what changes they have made to the requested questions.
According to the indication of the Journal we employed “Track changes”, the text highlighted in yellow is the method we used to make the revisions more evident. In the manuscript, we have included the revisions of both reviews as usual, in the last version you can see the second-round revisions of reviewer 1 highlighted in green. Moreover, many of your suggestions were similar to the other reviewer’s one, so it was impossible, in our opinion, to separate the revisions.

Round 3
Reviewer 2 Report
I appreciate the great effort the authors have made to respond to the entire request. First, given the large number of changes, in each reply to reviewer I would prefer that authors copy the text that has been recently incorporated into the manuscript, in addition to highlighting it in yellow.
In the first version of the abstract, there was a lack of definition of groups of study. Now, they are defined and assigned as Group_1, Group_2 and Group_3 in the methods section of abstract but this assignment is not used in the results section of the abstract. So, something seems unnecessary.
Later in the manuscript, in the Results section, lanes 107-110 there are mentions to Group_1 B, Group_2 B, Group_3 B, and Controls B which are not yet defined.
Please, note that the initial sentences in the results section are incomprehensible. What is the meaning of sentence in lanes 102-104 : “The markers investigated NADPH oxidase 2 (NOX2), 8-hydroxy-2′-deoxyguanosine 102 (8OHdG), nitrotyrosine (NT), nitric oxide synthase (iNOS) and Inteleukin 6 (IL-6), both for brain and placenta samples have been subjected a prelimary semiquantative analys and…”?
In lane 109 “the AQP4 was discarded like a possible marker” please change like to as.
Please, check for other typographical errors present in your manuscript.
From now I will be focused on scientific data. In methods section lane 343, the authors stated that only “one image for each organ (brain and placenta) and case of the different experimental groups was processed”. Whether this image is the one shown in the figure, I find very difficult to appreciate the differences in OS markers between controls and cases.
However, in the reply to my previous comment the authors mention that they analysed 341 slides. Please, add this information in the methods section, and indicate the number of analysed samples for each tissue and study group.
Please, could you show me the graphs in supplementary data as dot-plot instead of box-plot?
Previously I requested to indicate in these supplementary figures where the differences between groups are observed
Moreover, “Quantifications were expressed as number of positive-stained cells/analysed area”. For a defined area, the total number of cells will not always be the same, and therefore the number of positive cells. It will make the data very variable. Why did not the authors analyse data based on number of positive cells / number of total cells?
Author Response
Ferrara, July 15 2023
To the Editor of International Journal of Molecular Sciences
Dear Editor,
Please find enclosed a copy of our revised manuscript with the first, second and third round revisions “Oxidative stress markers in human brain and placenta may reveal the timing of hypoxic-ischemic injury: Evidence from an immunohistochemical study.” which is being submitted for publication in International Journal of Molecular Sciences-Special Issue “Molecular Mechanisms and Extracerebral Factors Affecting Brain Injury”.
We appreciate the third round of comments of Reviewer 2 and found them very useful to further improve the manuscript and better explain our study, which we know to be a very specific field of research for forensic pathology. We answered to their criticisms and incorporated changes in the appropriate sections of the revised manuscript (as Track changes different between the first and the second round of revisions). We also provided below a point-by-point reply to the Reviewer’ comments. We reduced the number of co-first authors to two, in the revised version of the manuscript.
The paper was been very improved and, we hope that now our manuscript, after the third round of revisions is now acceptable for publication in International Journal of Molecular Sciences.
Sincerely,
Margherita Neri, MD PhD
Department of Medical Sciences
University of Ferrara
Via Fossato di Mortara 70
44121 Ferrara, Italy
margherita.neri@unife.it
Point-by-point reply to the Reviewers’ comments
We appreciate the Reviewer’s third round comments and found them very helpful in further improving our manuscript and better clarifying our study. We revised our manuscript accordingly. Changes were fully incorporated in the appropriate sections of the revised manuscript and highlighted as “Track changes” in a different color, the revisions of the first and second rounds: first round revisions are highlighted in yellow; the second-round revisions are highlighted in green and the third round revisions are highlighted in light blue.
A point-by-point reply to the Reviewer’s comments is provided below.
Reviewer1 (R1):
I appreciate the great effort the authors have made to respond to the entire request. First, given the large number of changes, in each reply to reviewer I would prefer that authors copy the text that has been recently incorporated into the manuscript, in addition to highlighting it in yellow.
Thank you for appreciating our effort, in this third round we copied the text that has been recently incorporated into the manuscript, in addition to the highlighted text in the manuscript as “Track changes”.
In the first version of the abstract, there was a lack of definition of groups of study. Now, they are defined and assigned as Group_1, Group_2 and Group_3 in the methods section of abstract but this assignment is not used in the results section of the abstract. So, something seems unnecessary.
We improved the abstract by inserting the groups in the results, as you can see in the highlighted light blue text in the manuscript and in the following copy text:
“Results: The results for the brain samples suggest that NOX2, 8OHdG, NT, iNOS, and IL-6 were statistically significantly expressed in cases compared with Controls. iNOS is more expressed in Fetal intrauterine death (Group_1) and a little less expressed in Post-partum death (Group_3), while in Intra-partum death (Group_2) the immunoreactivity was very low. IL-6 showed in Fetal intrauterine death (Group_1) the highest expression in the brain cortex, while Intra-partum death (Group_2) and Post-partum death (Group_3) showed weak immunoreactivity. Post-partum death placentas (Group_3) showed the highest immunoreactivity to NOX2, which was almost double as compared to Fetal intrauterine death (Group_1) and Intra-partum death (Group_2) placentas. Placental tissues of Fetal intrauterine death (Group_1) and Intra-partum death (Group_2) showed higher expression of iNOS than Post-partum death (Group_3), while the IL-6 expression was higher in Fetal intrauterine death (Group_1) than Post-partum death (Group_3).”
Later in the manuscript, in the Results section, lanes 107-110 there are mentions to Group_1 B, Group_2 B, Group_3 B, and Controls B which are not yet defined.
Thank you for the suggestion, we insert the subgroups in the abstract, as you can see in the highlighted text in the manuscript and in the following copy text, because the format of the journal predicts that methods and materials are in the last part of the paper.
“Methods: A retrospective study was performed on the brains and placentas of fetuses and newborns, between 36-42 weeks of gestation (Group_1: Fetal intrauterine deaths, Group_2: Intrapartum deaths, Group_3: Post-partum deaths, Control group sudden neonatal death, all groups were further divided into two subgroups (Subgroup_B [brain] and Subgroup_P [placenta])”
Please, note that the initial sentences in the results section are incomprehensible. What is the meaning of sentence in lanes 102-104 : “The markers investigated NADPH oxidase 2 (NOX2), 8-hydroxy-2′-deoxyguanosine 102 (8OHdG), nitrotyrosine (NT), nitric oxide synthase (iNOS) and Inteleukin 6 (IL-6), both for brain and placenta samples have been subjected a prelimary semiquantative analys and…”?
We rephrase the text, to render clearer the paper, see line 106-110 of the manuscript, see the highlighted light blue text in the manuscript and in the following copy text:
“A preliminary semiquantitative analysis was performed, both for brain and placenta samples, the markers investigated were: NADPH oxidase 2 (NOX2), 8-hydroxy-2′-deoxyguanosine (8OHdG), nitrotyrosine (NT), nitric oxide synthase (iNOS), and Interleukin 6 (IL-6). Based on the results of the semi-quantitative analysis, the quantitative analysis was performed using ImageJ software (see section 4. Materials and Methods and the section 2.1 and 2.2 of Results ).”
In lane 109 “the AQP4 was discarded like a possible marker” please change like to as.
We changed like to as, see line 114 of the current version of the manuscript.
Please, check for other typographical errors present in your manuscript.
We performed a final check of the manuscript and correct the typographical errors. The changes are highlighted in light blue in the current version of the manuscript.
From now I will be focused on scientific data. In methods section lane 343, the authors stated that only “one image for each organ (brain and placenta) and case of the different experimental groups was processed”. Whether this image is the one shown in the figure, I find very difficult to appreciate the differences in OS markers between controls and cases.
After an evaluation of all database of images, we have welcomed your suggestion and have chosen more meaningful and representative images. See Figure 1 a,b and c; Figure 2 a and c; Figure 3 a,b and c, in the current version of the manuscript..
However, in the reply to my previous comment the authors mention that they analysed 341 slides. Please, add this information in the methods section, and indicate the number of analysed samples for each tissue and study group.
We add this information in the methods section, see lines 309-310 and lines 349-350 (the highlighted light blue text) of the current version of the manuscript, see the following copy text:
One sample of brain and placenta was collected for each case
One image for each organ (brain and placenta) and case of the different experimental groups was processed, a total of 341 slides was analyzed.
Please, could you show me the graphs in supplementary data as dot-plot instead of box-plot?
Previously I requested to indicate in these supplementary figures where the differences between groups are observed
We performed the dot plots of the data as you suggest, and we widely modified the supplementary materials adding the dot plots in the figures and integrating the legends. See the figures S1 c and d, S2 c and d, S3 c and d, S4 c and d, S5 c and d and the highlighted text in the respective legends. As you can see the supplementary is very improved now, and your suggestions were very useful.
Moreover, “Quantifications were expressed as number of positive-stained cells/analysed area”. For a defined area, the total number of cells will not always be the same, and therefore the number of positive cells. It will make the data very variable. Why did not the authors analyse data based on number of positive cells / number of total cells?
The methodology of the study was to see all samples, the semiquantitative analysis was performed on all the slice areas, for each case, and then we selected the more significative area for each sample and acquire the image for the quantitative analysis. For more correct results we prefer to perform the quantification as the number of positive-stained cells/analysed area, because it was very difficult to have the same number of cells in each sample, so we acquired images with a comparable number of cells between samples.
We really appreciate that your suggested revisions to the paper have made the manuscript better.
We thank you for your further revisions, we perform the changes in the manuscript, you can see the third round track changes highlighted in light blue in the text.

Round 4
Reviewer 2 Report
The aim of the authors is to present a scientific document on which forensic doctors can find legal support. Here, this scientific document is based on 25 cases splatted into 3 groups. For each case, the study took one sample of brain and one sample of placenta. A total of 341 slides were analysed, we still do not know the number of slides per organ and group. Eventually, different markers were analysed. We do not know the number of slides analysed per marker.
I assume based on the author’s statement “One image for each organ (brain and placenta) and case of the different experimental groups was processed” that only one slide per marker, organ and case was analysed. With only 1 slide, not even analysing 2 or more slides of the same organ for every marker, I found difficult to present this scientific document as liable, mainly because the data from cases show high variability.
Another worrying issue of variability is that the authors use a questionable method of counting number of positive cells per area. I had proposed counting number of positive cells per total number of cells. Their answer is “For more correct results we prefer to perform the quantification as the number of positive-stained cells/analysed area, because it was very difficult to have the same number of cells in each sample”. First, the use of the total number of cells in the denominator is to normalize the data. And second, the authors acknowledge the variability in total number of cells and, consequently, the variability in the number of positive cells.
As secondary comments to the last version of the manuscript:
In supplementary figures, why is the scale of the y-axis scale different between the dot-plots and the box-plots?
In figure S1b, authors reported statistical differences between group 1 and Group 2. In view of box-plot and dot-plot graphs, and given the low n, I found it very difficult to observe differences between these two groups.
Author Response
The answer to the reviewer is enclosed in the attached file.

Round 5
Reviewer 2 Report
I appreciate the authors's efforts' to improve and correct the manuscript. However, there are some points to clarify
In the last upload of supplemental material, legend for figure S1b reads:
S1b. Kruskal-Wallis test: p<0.05; Mann-Whitney U test: Group 3 vs controls: p<0.05; Group 1 vs Group 2: p<0.05
Authors reported statistical differences in the NOX2 expression between placentas of group 1 and Group 2 (supplementary Figure S1b and S1d). It is very hard to believe that these two groups show differences by carefully observing box plot and dot-plot of these figures.
